∂ | **Open Peer Review** | Bacteriology | Research Article

# SarS and Rot are necessary for the repression of *lukED* and *lukSF-PV* in *Staphylococcus aureus*

Exene E. Anderson,[1] Juliana K. Ilmain,[1] Victor J. Torres[1,2]

**ABSTRACT** *Staphylococcus aureus* is an opportunistic pathogen that employs an array of different virulence factors to evade the immune system. The bi-component pore-forming leukocidins are of particular importance for their ability to target and kill phagocytes. The operon of two of these toxins, *lukED* and *lukSF-PV*, are repressed by two proteins, SarS and Rot. However, how these repressors work together to repress these toxins is not completely understood. Here, we determine that repression of leukocidins by SarS and Rot is not additive and that SarS and Rot are able to bind concurrently to the leukocidin promoters. In addition, in a tissue culture model using primary human neutrophils, the deletion of both repressors did not result in increased virulence compared to the single deletions of *sarS* and *rot*. Further experiments revealed that in an *in vivo* mouse infection model, virulence was similar in strains lacking one versus both repressors. Overall, these data show that while the repression of leukocidins by SarS and Rot is not additive, both proteins are critical for the repression of these toxins.

**IMPORTANCE** The leukocidins play an important role in disarming the host immune system and promoting infection. While both SarS and Rot have been established as repressors of leukocidins, the importance of each repressor in infection is unclear. Here, we demonstrate that repression by SarS and Rot is not additive and show that in addition to upregulating expression of each other, they are also able to bind concurrently to the leukocidin promoters. These findings suggest that both repressors are necessary for maximal repression of *lukED* and *lukSF-PV* and illuminate another complex relationship among *Staphylococcus aureus* virulence regulators.

**KEYWORDS** MRSA, toxin, leukocidins, regulation, infection, transcriptional repression, Rot, SarS, LukED, PVL, LukSF

*S*taphylococcus aureus is a remarkable pathogen that can infect a wide range of tissues and cause a variety of different ailments, from superficial skin and soft tissue infections to life-threatening pneumonia and sepsis (1). *S. aureus* produces a number of different virulence factors to aid in the infection of the host, including the bi-component pore-forming leukocidins (2). These toxins, which include LukAB (also known as LukHG), LukED, LukSF-PV (also known as PVL), HlgAB, and HlgCB, disable host immune response by lysing a variety of host immune cells, including neutrophils, monocytes, macrophages, and T cells (3).

The production of leukocidins is overarchingly regulated by the accessory gene regulatory (Agr) system. The Agr system responds to quorum sensing to regulate a wide range of virulence factors (4). The activation of the Agr system *in vitro* leads to an increase in the production of exotoxins and secreted proteins and a decrease in the production of cell-wall-associated proteins (5). This regulation is performed by the effector molecule of the Agr system, RNAIII (6, 7). Two targets of RNAIII are SarS and Rot (repressor of toxins) (8–10). Both proteins repress the leukocidins: Rot broadly represses all the toxins (11,

Address correspondence to Victor J. Torres, victor.torres@stjude.org.

V.J.T. is an inventor on patents and patent applications filed by New York University, which are currently under commercial license to Janssen Biotech, Inc. Janssen Biotech, Inc., provides research funding and other payments associated with the licensing agreement. All other authors declare no conflicts of interest.

See the funding table on p. 10.

12), while SarS selectively represses *lukED*, *lukSF-PV*, and *hla* (8, 9, 13). Both Rot and SarS can repress leukocidins by directly binding to the promoter region of toxin genes and inhibiting their expression (9, 13, 14). In addition, Rot and SarS are activators of each other, working in an epistatic relationship (13, 15). Beyond being repressed by RNAIII and activators of each other, SarS and Rot are entangled in the complex regulatory network of *S. aureus*. *sarS* is additionally activated by SarT (16), TcaR (17), and post-transcriptionally by *gdpS* (18), while it is repressed by SarA (8, 9) and MgrA (19). Rot has also been shown to repress toxins through the inhibition of the two-component SaeRS system (14, 20), which is a critical activator of leukocidin expression (21–25).

Rot works as a dimer, where each monomer contains a winged helix-turn-helix DNA-binding motif (12, 26). It has been shown that Rot preferentially recognizes AT-rich regions of DNA and that this repressor is most likely to bind to a region of DNA that is between 18 and 30 nucleotides (12, 26). SarS, which, like Rot, is part of the SarA family of proteins, differs from the other members of that family in that it is not a dimer, but instead, it is a larger protein that contains two homologous halves (27). SarS contains two winged-helix DNA-binding domains and has folds that are similar to those found in SarR and MarR. For both SarS and Rot, it has been suggested that the helix-turn-helix motifs interact with the major grooves of DNA, while the winged motifs interact with the minor grooves (12, 27).

While genetic experiments demonstrate that Rot and SarS are both regulating each other as well as regulating the leukocidins, it is not fully understood how this regulation occurs and how this affects the virulence of *S. aureus*. Two outstanding questions that we aimed to explore are (i) whether the repressors are additive in their repression of toxin expression and (ii) whether they are both necessary for repression *in vivo*. Gaining a better understanding of how leukocidins are regulated will allow us to better understand how *S. aureus* tightly controls virulence factors and how it is such an adaptable pathogen.

In this study, we aimed to further explore the relationship between SarS and Rot, as well as delve into how these transcription factors are repressing the leukocidins. We determined that SarS and Rot are not additive in their repression of leukocidins at the transcript or protein level and demonstrated that deletion of both *sarS* and *rot* does not alter the pathogenesis of *S. aureus* in tissue culture models of infection and a murine model of infection, as compared to the single deletion strains. In addition, we showed that SarS and Rot can bind concurrently to the leukocidin promoters. Together, these data demonstrate that SarS and Rot are both necessary for the repression of leukocidins.

## RESULTS

### SarS and Rot are not additive in their repression of leukocidin expression

While it had been established that both SarS and Rot can repress *lukED*, *lukSF-PV,* and *hla* (9, 11–13), it was unclear whether SarS and Rot were additives in their repression of leukocidins. To begin answering this question, we looked at transcript levels of *lukED*, *lukSF-PV,* and *hla* in the *ΔsarS, Δrot,* and *ΔsarSΔrot* strains as compared to the wild-type strain. For these studies, we used strain AH-LAC, a USA300 community-associated methicillin-resistant *S. aureus* (CA-MRSA) strain. We observed no significant increase in *lukSF-PV* or *hla* transcript level between the single deletion strains and the double deletion strain at either exponential (Fig. 1A) or stationary phase (Fig. 1B). There was increased upregulation of *lukED* in the double deletion strain as compared to the *sarS* deletion strain at both 3 and 5 hours, but this was a *lukED*-specific phenotype. However, there was no significant difference in the upregulation of *lukED* between the *ΔsarSΔrot* strain and the *Δrot* strain at either time point. We also noted that overall, the deletion of *rot* led to a modest increase in *lukED* and *lukSF-PV* transcript levels than the deletion of *sarS*, suggesting that the Rot may be slightly dominant over SarS. Overall, these data suggest that SarS and Rot are not additive in their repression of leukocidin transcription.

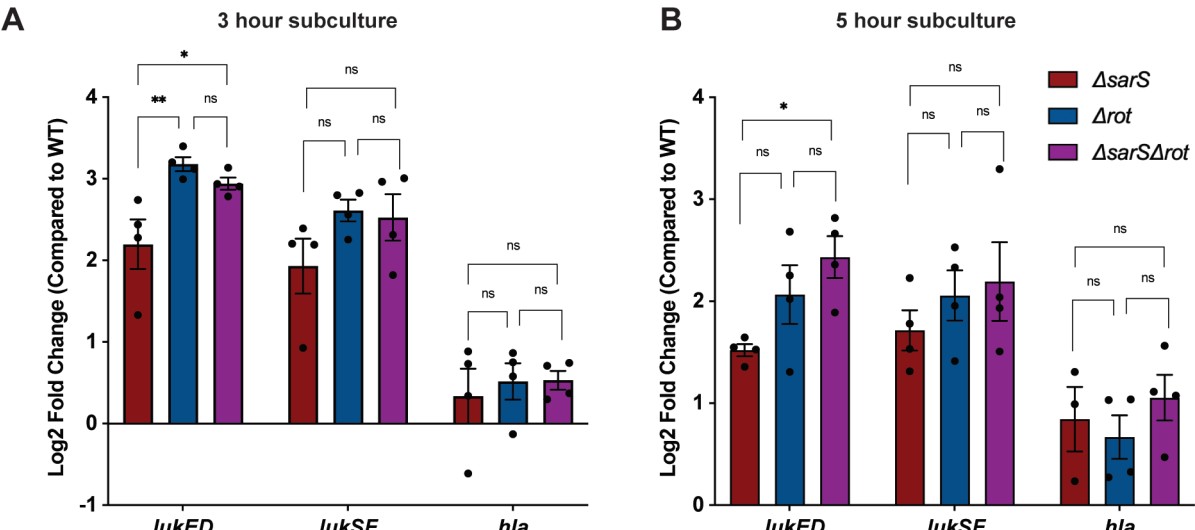

**FIG 1** Deletion of both *sarS* and *rot* does not result in accumulative upregulation of leukocidin transcripts. Log2 fold change of pore-forming toxin transcript levels was measured by qRT-PCR in the *ΔsarS, Δrot,* and *ΔsarSΔrot* strains and compared to wild-type AH-LAC in both exponential (A) and stationary (B) phases of growth ($n = 4$). Statistical analysis was performed using a one-way ANOVA with multiple comparisons, and error bars are showing the standard error of the mean. *P*-values for this figure are as follows: *$P < 0.05$, **$P < 0.01$.

## SarS and Rot are not additive in their repression of leukocidin production

We next investigated the exoprotein profiles of isogenic wild-type, *ΔsarS, Δrot,* and *ΔsarSΔrot* strains to determine whether deleting both transcription factors had a synergistic effect. We observed that deletion of *sarS* and *rot* resulted in very similar, but not identical, changes in exoprotein profiles as compared to wild type (Fig. 2A). In addition, we noticed increased abundance of proteins that corresponded to the size of the leukocidins (~35 kDa and ~43 kDa), but we did not observe notable difference in abundance between the single and double deletion strains.

We also performed western blots using toxin-specific antibodies to determine the impact that deleting *sarS* and *rot* had on leukocidin production. We observed higher levels of LukE, LukF-PV, and α-toxin at both 3 and 5 hours in the *ΔsarS, Δrot,* and *ΔsarSΔrot* strains as compared to the wild type (Fig. 2B). Notably, we did not see an accumulative increase in the *ΔsarSΔrot* strain as compared to the *ΔsarS* or *Δrot* strains. We noted that deletion of *rot* resulted in higher production of toxins than deletion of *sarS*, further supporting the observation that Rot may be slightly more dominant than SarS.

## *ΔsarSΔrot* is not more virulent than *ΔsarS* or *Δrot*

To investigate whether deleting both repressors had an additive effect on virulence, we measured the cytotoxicity of supernatants from the wild-type, *ΔsarS, Δrot,* and *ΔsarSΔrot* USA300 strains on primary human neutrophils. We observed that supernatants from all three of the deletion strains killed significantly more neutrophils than the wild-type strains (Fig. 3A). However, there was no difference in killing between the supernatants from the *ΔsarS, Δrot,* or *ΔsarSΔrot* strains. To further test the virulence of these strains, we infected neutrophils with live bacteria and measured neutrophil-mediated lysis by the bacteria. Similar to the intoxication studies, we observed significantly more killing of neutrophils by the strains lacking *sarS*, *rot*, or both repressors, but not an accumulative increase in the strain lacking both toxins as compared to the *Δrot* strain (Fig. 3B). However, the *ΔsarSΔrot* strain did kill more neutrophils than the *ΔsarS* strain. This correlates with the leukocidin transcript levels observed in the deletion strains, where Rot is able to minimally repress leukocidins in the absence of SarS.

To further define the role of these repressors in pathogenesis, we utilized an *in vivo* murine intraperitoneal systemic infection model. Mice were infected with the wild-type,

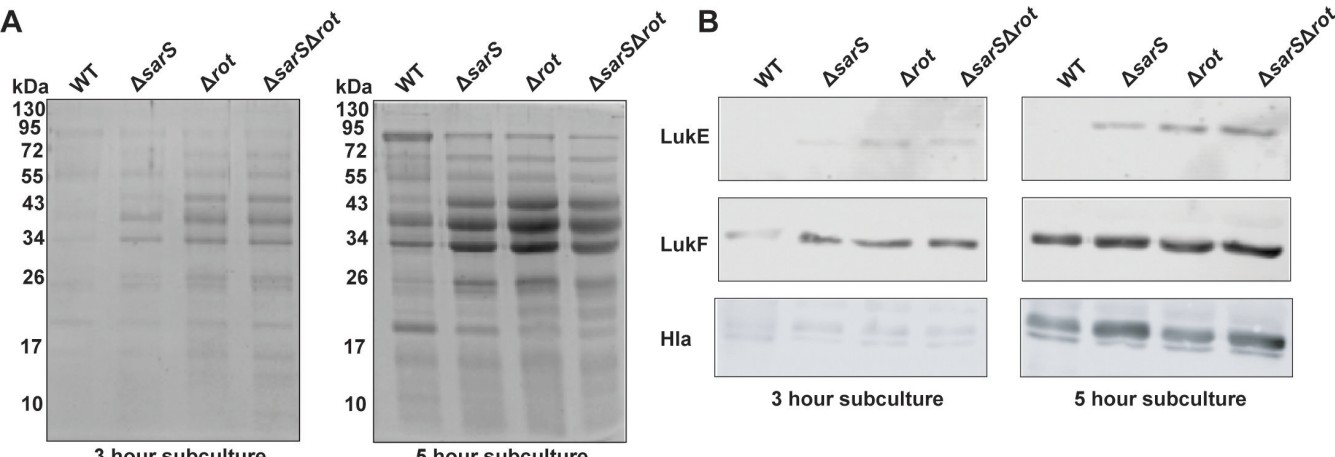

**FIG 2** Deletion of both *sarS* and *rot* does not result in an accumulative increase in leukocidin production. (A) Supernatants collected from optical-density-normalized wild-type AH-LAC, Δ*sarS*, Δ*rot*, and Δ*sarS*Δ*rot* strains in exponential phase (3 hours) and stationary phase (5 hours) show altered exoprotein profiles. Representative Coomassie-stained gels are shown. (B) Immunoblots of samples from panel A measuring LukE, LukF-PV, and α-toxin production at both exponential and stationary phases of growth. Representative immunoblots are shown.

Δ*sarS*, Δ*rot*, and Δ*sarS*Δ*rot* USA300 strains and monitored for morbidity and mortality. Significantly, more mice infected with the Δ*sarS*, Δ*rot*, and Δ*sarS*Δ*rot* strains succumbed to the infection as compared to the mice infected with the wild-type strain (Fig. 3C). However, the strain lacking both repressors was as virulent and not significantly more than the Δ*sarS* or Δ*rot* strains. Altogether, these data demonstrate that deleting both repressors does not increase the virulence of USA300 as compared to the deletion of either single repressor.

## SarS and Rot can bind concurrently on leukocidin promoters

Having determined that SarS and Rot are not additive in their repression of the leukocidins, we next wanted to gain more insight into how these two transcription factors repress the toxins. We hypothesized that SarS and Rot can occupy the leukocidin promoter concurrently. To test this, we performed a promoter pull-down assay where biotinylated DNA probes were conjugated to streptavidin magnetic beads, which were used to capture His-tagged SarS, His-tagged Rot, or both proteins. Binding was then detected *via* immunoblotting. We used the *purA* promoter as a negative control, as the literature suggested both SarS and Rot should not bind to this promoter (8, 9, 15). We observed that both SarS and Rot bound to the P*lukSF* and P*lukED* promoters individually as well as when both regulators were combined, while no binding was detected with the P*purA* (Fig. 4A). From these data, we can conclude that SarS and Rot are able to occupy the leukocidin promoters at the same time.

The observation that the two repressors can bind concurrently to the toxin promoters suggested that they may both be necessary for maximal repression of the leukocidins. This idea was strengthened by previous research that had shown Rot to be an activator of *sarS* in a derivative of *S. aureus* strain NCTC 8325 that was cured of all prophages (15), and SarS to be an activator of *rot* promoter activity and protein production in USA300 (13). To validate the regulatory relationship between *sarS* and *rot* in the USA300 strains used in this study, we performed qRT-PCR to measure transcript levels of *sarS* and *rot* in a Δ*rot* and Δ*sarS* strain, respectively. We observed that there was a decrease in *sarS* transcript in the Δ*rot* strain (Fig. 4B), and a similar decrease in *rot* transcript in the Δ*sarS* strain (Fig. 4C), at both exponential (3 hours) and stationary (5 hours) phases of growth. These data establish that SarS and Rot are activators of each other in USA300 and support the idea that both are needed for full repression of *lukED* and *lukSF-PV*.

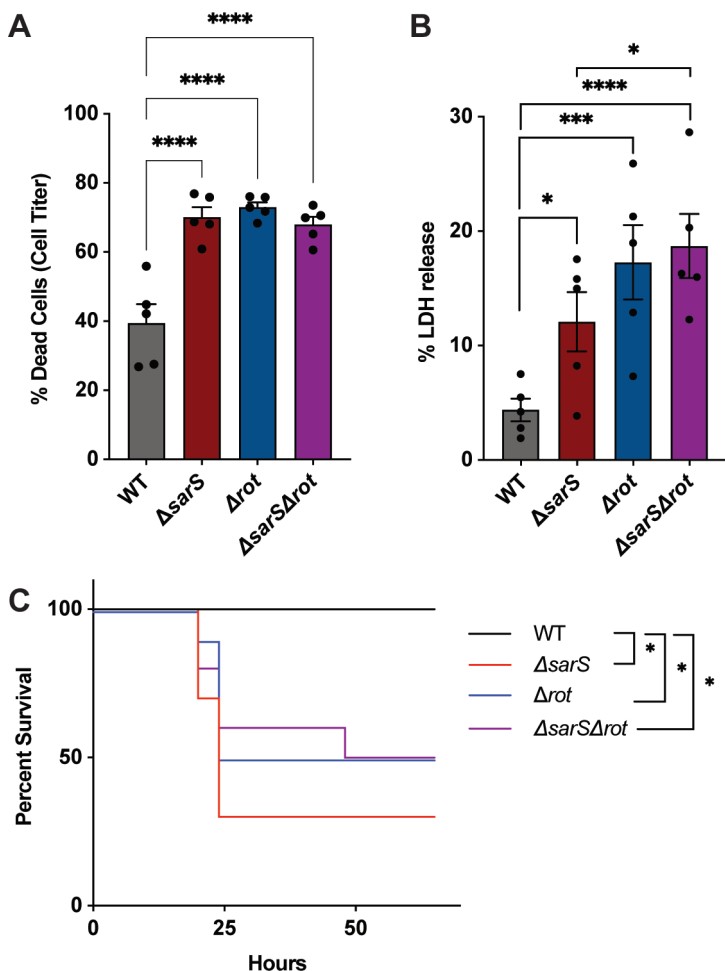

FIG 3   Deletion of both *sarS* and *rot* does not result in a more virulent strain than *ΔsarS* or *Δrot*. (A) Culture supernatants collected from the *ΔsarS, Δrot,* and *ΔsarSΔrot* strains (5%) are more cytotoxic toward primary human polymorphonuclear neutrophils (PMNs) than supernatants collected from the wild-type AH-LAC strain. Cell death of the PMNs was measured with the Cell Titer. Each data point represents an individual donor (*n* = 5). (B) Primary human PMNs infected with wild-type AH-LAC, *ΔsarS, Δrot,* and *ΔsarSΔrot* strains at a multiplicity of infection of 6.25 are more susceptible to the *ΔsarS, Δrot,* and *ΔsarSΔrot* strains than the wild-type strain, and more susceptible to the *Δrot* and *ΔsarSΔrot* strains than the *ΔsarS* strain. Cell death of the PMNs was measured with lactate dehydrogenase release. Each data point represents an individual donor (*n* = 4). (A–B) Statistical analysis was performed using a two-way ANOVA with multiple comparisons, and error bars are showing the standard error of the mean. (C) 8-week-old B6 female mice were infected by intraperitoneal injection with wild-type AH-LAC, *ΔsarS, Δrot,* and *ΔsarSΔrot S. aureus* at a concentration of $1 \times 10^8$ CFU/mouse and monitored for survival (two independent experiments, *n* = 10 mice per group). Statistical analysis was performed using the Log-rank Mantel-Cox test. *P*-values for this figure are as follows: *$P < 0.05$, ***$P < 0.001$, ****$P < 0.0001$.

## DISCUSSION

While the bi-component pore-forming leukocidins are a critical component of virulence in *S. aureus*, there are still gaps in our understanding of how these toxins are regulated. In this work, we described our investigation of the repression of *lukED* and *lukSF-PV* by SarS and Rot. We demonstrated that SarS and Rot are not additive in their repression of toxins (Fig. 1 and 2). In addition, tissue culture models using primary human neutrophils and an *in vivo* infection murine model demonstrated that deletion of both repressors does not result in a more virulent strain as compared to deletion of a single repressor (Fig. 3). Finally, we showed that SarS and Rot can bind concurrently to leukocidin promoters and

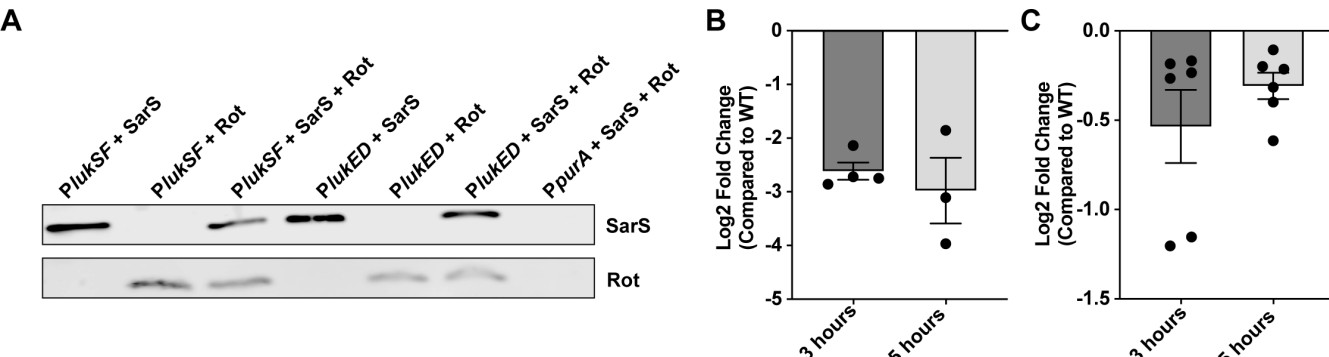

**FIG 4** SarS and Rot can occupy leukocidin promoters concurrently. (A) Biotinylated promoter DNA was incubated with His-tagged SarS, His-tagged Rot, or both proteins. Protein binding was visualized by immunoblotting. Representative images are shown. (B) Log2 fold change of *sarS* transcript levels in the *Δrot* strain compared to the wild-type USA300 strain at exponential and stationary phases (*n* = 3–4). (C) Log2 fold change of *rot* transcript levels in the *ΔsarS* strain compared to the wild-type USA300 strain at exponential and stationary phases (*n* = 3).

confirmed that the two repressors are activators of each other in the CA-MRSA strain USA300 (Fig. 4).

The observation that SarS and Rot are not additive in their repression of toxins and that they can occupy the leukocidin promoter at the same time suggests that these repressors may both be necessary for the maximal repression of toxins. While we did observe a slight increase in toxin transcript levels in the *Δrot* and *ΔsarSΔrot* strains as compared to the *ΔsarS* strain, there was no difference in transcript levels between the *Δrot* and *ΔsarSΔrot* strains. We hypothesize that both repressors are needed to fully repress the toxins. Rot and SarS activating each other in an epistatic relationship further strengthens the idea that both are necessary for repression. We posit that either both proteins need to bind to the leukocidin promoter for full repression of the toxins or that when one protein is not present, it is unable to induce the production of the other protein, therefore leading to a lack of repression of the leukocidins. Further research is needed to fully elucidate the relationship between these repressors and the leukocidins.

Rot has previously been shown to work cooperatively with other regulators. Rot and SaeR are able to work synergistically to activate the expression of superantigen-like exoproteins (21). In addition, modeling has suggested that SarS and Rot also act cooperatively in the activation of the *spa* (28). Therefore, it is feasible that Rot and SarS are cooperating to repress the leukocidins. Additional work is needed to uncover the molecular means of these interactions.

While the importance of Rot and SarS to the regulation of virulence factors in *S. aureus* is well established (13, 14, 29, 30), the binding sites for Rot or SarS have yet to be elucidated. Understanding where these regulators bind will expand our understanding of how these repressors work. It could also shed light not only on how SarS and Rot interact with each other but also on how they interact with other regulators of leukocidins such as SaeR.

Altogether, the findings presented here continue to demonstrate how complex toxin regulation is in *S. aureus*, and additional studies are needed to shed light on the intricacies of the regulatory network and the interplay between regulators. However, it is clear that toxin expression is highly regulated in *S. aureus*. Necessitating two repressors to inhibit the leukocidins may be one way in which the bacterium tightly controls the expression of lethal toxins.

## MATERIALS AND METHODS

### Bacterial growth conditions

*S. aureus* strains (Table 1) were routinely grown at 37°C on tryptic soy agar or in tryptone soya broth (TSB). *Escherichia coli* bacteria were grown in Luria-Bertani broth. Liquid cultures were grown in 5 mL of growth medium in 15 mL conical tubes and incubated at a 45° angle with shaking at 180 rpm unless otherwise specified. For all experiments involving the growth of *S. aureus*, a 1:100 dilution of overnight cultures was subcultured into a fresh medium.

### Construction of bacterial strains and plasmids

For all the strains and oligonucleotides used in this study, see Tables 1 and 2.

The *ΔsarSΔrot* strain was constructed by phage transduction using phage 80α lysate from the *ΔsarS::ermC* strain into the *Δrot::spec* strain and was selected for with 5 µg/mL erythromycin.

### RNA isolation

Bacteria were subcultured for 3 and 5 hours in 5 mL of TSB before being spun down and resuspended in 1 mL of RNA Stat-60 (Amsbio). Samples were bead beaten in Lysing Matrix B tubes (MP Bio) using the FastPrep and spun down for 10 minutes at 12,000 *g*. The upper layer was collected and 200 µL of chloroform was added. The samples were incubated at room temperature for 3 minutes before being spun down for 15 minutes at 12,000 *g*. The aqueous phase was removed, and 0.5 mL of isopropanol was added to precipitate RNA. RNA was washed twice with 70% ethanol, air-dried, and resuspended in RNase-free water. RNA (10,000 ng) was DNase treated (TURBO DNA-free kit, Invitrogen Ambion).

### qRT-PCR

DNase-treated samples were converted into cDNA with the SuperScript III First-Strand Synthesis Kit (Thermo Scientific). Next, 1 µL of cDNA was added to TaqMan probes and Universal PCR Master Mix. Transcript levels were measured using the QuantStudio 3 System. Genes were normalized to the *rpoB* housekeeping gene and reported as $2^{-\Delta CT}$. The probe used was as follows: PrimerTime 5′ Hex/ 3′ BHQ-1.

### Exoprotein isolation, Coomassie staining, and immunoblotting

The proteins in the culture supernatants of bacteria grown for either 3 or 5 hours were precipitated and analyzed as previously described by Benson et al (14). Briefly, following optical density (OD) normalization, the cultures were spun down, and 1.4 mL of supernatant was added to 140 µL of trichloroacetic acid (TCA) and left overnight at 4°C. The precipitated proteins were sedimented, washed, dried, resuspended in 8M urea, and left at room temperature for 30 minutes. Next, 2× SDS loading buffer was added and the mixture was boiled for 10 minutes. Proteins were separated on a 12% SDS-PAGE gel, transferred to nitrocellulose membranes, and probed with indicated primary antibodies.

**TABLE 1**   Bacterial strains used in this study

| VJT # | Strain name | Description | Reference |
|---|---|---|---|
| 15.77 | AH-LAC | AH1263 Erm[S] USA300 parent strain | (31) |
| 84.97 | AH-LAC *ΔsarS* | AH-LAC containing an erythromycin cassette insertion into *sarS* (this strain is erythromycin resistant) | (13) |
| 17.37 | AH-LAC *Δrot* | AH-LAC containing a spectinomycin cassette insertion into *rot* (this strain is spectinomycin resistant) | (32) |
| 84.99 | AH-LAC *ΔsarSΔrot* | AH-LAC containing erythromycin cassette insertion into *sarS* and spectinomycin cassette insertion into *rot* (this strain is erythromycin and spectinomycin resistant) | This study |

**TABLE 2** Oligonucleotides used in this study

| VJT # | Description | Sequence |
|---|---|---|
| 2439 | rpoB_F | 5′-GAACATGCAACGTCAAGCAG |
| 2440 | rpoB_R | 5′-AATAGCCGCACCAGAATCAC |
| 2740 | rpoB_probe | 5′-TACAGGTATGGAACACGTTGCAGCA |
| 2734 | lukE_F | 5′-GGACTGACGACTAAAGATCCAAA |
| 2735 | lukE_R | 5′-AATGAGCCATTGCCACCTAT |
| 2736 | lukE_probe | 5′-TGGAGGTAATTTCCAGTCAGCACCA |
| 2435 | lukS_F | 5′-GCTGCAACATTGTCGTTAGG |
| 2436 | lukS_R | 5′-GCGCCATCACCAATATTCTC |
| 2737 | lukS_probe | 5′-TCACTCCTATTGCTACTTCGTTTCATG |
| 3015 | hla_F | 5′-AGATTCTTGGAACCCGGTATATG |
| 3016 | hla_R | 5′-CTGTAGCGAAGTCTGGTGAAA |
| 3011 | hla_probe | 5′-TGGCTCTATGAAAGCAGCAGATAACTTCC |
| 3404 | sarS_F | 5′-CAATCCACCATAAATACCCTCAAAC |
| 3405 | sarS_R | 5′-GCTGCGCGTCATCCATA |
| 3406 | sarS_probe | 5′-AGAACGCTCAACTGAAGATGAAAGA |
| 3214 | PlukE_F | 5′-CTTATTTGAAAAAAGCAAAAAAGATAGG |
| 3215 | PlukE_R_B | 5′-CTTAAACATAAGTTTCACTTTCTTTC |
| 2306 | PpurA_F | 5′-AAAAGTTTTTCCGTACAATA |
| 2307 | PpurA_R_B | 5′-ACATGTGAGCACCTCCAAGT |

Immunoblotting was performed with monoclonal antibodies against LukE (1:5,000, Envigo antibody 1–31.3.3.3) and LukF-PV (1:5,000, Envigo antibody 2–4.2.6), which were detected with a fluorescent Alexa Fluor 680-conjugated anti-mouse antibody (1:25,000, Invitrogen). α-toxin (1:5,000) was detected with a polyclonal antibody (Sigma) and a fluorescent Alexa Fluor 680-conjugated anti-rabbit antibody (1:25,000; Invitrogen).

## Cytotoxicity assay and extracellular human polymorphonuclear neutrophil infections

Human polymorphonuclear neutrophils (hPMNs) were isolated using a Ficoll-Paque method as described before (33). Briefly, PMNs were seeded at $2 \times 10^5$ cells per well in Roswell Park Memorial Institute media (without phenol red; Fisher Scientific) supplemented with 10% heat-inactivated fetal bovine serum (Gemini BioProducts). Either 1.25% or 2.5% supernatant was added (percentage indicates the percentage of total reaction volume that was culture supernatant). To measure cell viability, the metabolic dye CellTiter (Promega) was added at a final concentration of 10% per well and incubated for 2 hours at 37°C + 5% $CO_2$. Absorbance at 492 nm was measured using the PerkinElmer EnVison plate reader. For extracellular infections, bacteria were subcultured for 3 hours in 5 mL of TSB, washed twice with PBS, and normalized to an $OD_{600}$ of 1. PMNs were added to a flat-bottomed tissue culture-treated plate at $2 \times 10^5$ and incubated at room temperature for 30 minutes. Bacteria were added at a multiplicity of infection of 100. After a 2-hour infection at 37°C and 5% $CO_2$, PMN viability was determined by lactate dehydrogenase release (CytoTox-ONE homogenous Membrane Integrity Assay, Promega), measured using the PerkinElmer EnVison plate reader.

## Animal housing conditions

Animals received PicoLab Rodent Diet 20 (LabDiet) and acidified water, and were housed under normal lighting cycle conditions (12 hours ON/12 hours OFF) and a temperature of 70°F.

## Murine intraperitoneal infection

Bacteria were subcultured for 3 hours in 5 mL of TSB, washed twice in PBS, and OD normalized. 8-week-old C57BL/6J female mice (Jackson Laboratory) were injected with

200 µL of bacteria intraperitoneally. Mice were monitored multiple times a day for 72 hours and euthanized upon severe signs of mortality or excessive weight loss.

## SarS purification

Full-length *sarS* was amplified from AH-LAC, adding XhoI and BamHI cut sites and cloned into pET15b. The plasmid was transformed into BL21-DE3g. The expression strain was grown in 400 mL LB broth with 100 µg/mL ampicillin at 37°C and 250 rpm until the cultures reached an $OD_{600}$ of 0.6. The culture was induced with 1 mM isopropyl ß-D-1-thiogalactopyranoside (IPTG) and grown for 4 hours at 37°C 250 rpm. The culture was spun down and resuspended in 20 mM Tris pH 7.5, 300 mM NaCl, and 10% glycerol. Cells were lysed in 1× protease inhibitor (Pierce Protease inhibitor cocktail) and sonicated on ice. Bugbuster (Millipore) at 1× was added, and the cultures were incubated at room temperature for 35 minutes. Following incubation on ice for 15 minutes and centrifugation, the supernatant was filtered through a 0.2-µm filter. The protein was purified using a HisTrap HP column on an AKTA pure chromatography system (Cytiva), eluting with a linear gradient in elution buffer (20 mM $Na_2HPO_4$, 500 mM NaCl, 400 mM Imidazole, and pH 7.4). Purified protein was dialyzed in 10% glycerol in TSB.

## Rot purification

Full-length Rot was amplified from AH-LAC, adding XhoI and NdeI cut sites and cloned into pET14b. The plasmid was transformed into BL21-DE3g. The expression strain was grown in 400 mL LB broth with 100 µg/mL ampicillin at 37°C and 250 rpm until the cultures reached an $OD_{600}$ of 0.6. The culture was induced with 1 mM IPTG and grown for 4 hours at 37°C 250 rpm. The culture was spun down and resuspended in 20 mM Tris pH 7.5, 300 mM NaCl, and 10% glycerol. Cells were lysed in 1× protease inhibitor (Pierce Protease inhibitor cocktail) and sonicated on ice. Bugbuster (Millipore) at 1× was added, and the cultures were incubated at room temperature for 35 minutes. Following incubation on ice for 15 minutes and centrifugation, the supernatant was filtered through a 0.2-µm filter. The protein was purified using a HisTrap HP column on an AKTA pure chromatography system (Cytiva), eluting with a linear gradient in elution buffer (20 mM $Na_2HPO_4$, 500 mM NaCl, 400 mM imidazole, and pH 7.4). Purified protein was dialyzed in 10% glycerol in TSB.

## Promoter pull-down assays

We generated PCR products for each promoter using oligonucleotides containing biotinylated labels and purified the PCR products (Qiagen). M-280 streptavidin Dynabeads (Invitrogen, 11205D) were washed and resuspended in wash buffer [2M NaCl, 1 mM ethylenediaminetetraacetic acid (EDTA), 10 mM Tris, and pH 7.5]. The beads were incubated with 800 fmol of each DNA fragment for 30 minutes at room temperature on a rotisserie. The samples were washed three times with wash buffer, before being resuspended in binding buffer (25 mM Tris-HCl, 0.1 mM EDTA, 75 mM NaCl, 10% glycerol, 1 mM dithiothreitol [DTT], pH 7.5). SarS or Rot at a concentration of 100 nM and 5 µg of poly(dG:dC) was added to the beads and mixed on a shaking platform for 15 minutes at 30°C and 550 rpm. Following incubation, beads were washed twice with binding buffer, resuspended in 1× SDS sample buffer, and boiled for 10 minutes. Following electrophoresis, gels were transferred to a nitrocellulose membrane and probed with an anti-His antibody (1:2,000, Cell Sciences, CSI20563B) and detected with a fluorescent Alexa Fluor 680-conjugated anti-mouse antibody (1:25,000, Invitrogen).

## Statistical methods

Prism software (GraphPad, Inc.) was used to perform statistical analysis. One-way ANOVA was utilized for qRT-PCR, cytotoxicity, and hPMN infection. The statistical significance of the difference between survival curves was determined by the Log-rank Mantel-Cox test.

## ACKNOWLEDGMENTS

We thank members of the Torres laboratory for their insightful discussions and comments on this manuscript.

This work was supported in part by the NIH-National Institute of Allergy and Infectious Diseases award numbers R01s AI105129 and AI137336 and by ALSAC (V.J.T.).

E.E.A. and V.J.T. designed the study; E.E.A. performed all the experiments; J.K.I. and E.E.A. generated the SarS and Rot proteins; E.E.A. and V.J.T. wrote the manuscript; and J.K.I. commented on the manuscript.

## AUTHOR AFFILIATIONS

[1]Department of Microbiology, New York University Grossman School of Medicine, New York, New York, USA

[2]Department of Host-Microbe Interactions, St. Jude Children's Research Hospital, Memphis, Tennessee, USA

## AUTHOR ORCIDs

Exene E. Anderson  http://orcid.org/0009-0006-2437-7445
Victor J. Torres  http://orcid.org/0000-0002-7126-0489

## FUNDING

| Funder | Grant(s) | Author(s) |
| --- | --- | --- |
| HHS \| NIH \| National Institute of Allergy and Infectious Diseases (NIAID) | AI105129 | Victor J. Torres |
| HHS \| NIH \| National Institute of Allergy and Infectious Diseases (NIAID) | AI137336 | Victor J. Torres |

## AUTHOR CONTRIBUTIONS

Exene E. Anderson, Conceptualization, Data curation, Formal analysis, Investigation, Methodology, Writing – original draft, Writing – review and editing | Juliana K. Ilmain, Methodology, Writing – review and editing | Victor J. Torres, Conceptualization, Funding acquisition, Project administration, Supervision, Writing – review and editing

## ETHICS APPROVAL

Buffy coats were obtained from anonymous blood donors with informed consent from the New York Blood Center. All animal experiments were reviewed and approved by the Institutional Animal Care and Use Committee of New York University Langone Medical Center (NYULMC). All experiments were performed according to NIH guidelines, the Animal Welfare Act, and U.S. federal law.

## ADDITIONAL FILES

The following material is available online.

Open Peer Review

**PEER REVIEW HISTORY (review-history.pdf).** An accounting of the reviewer comments and feedback.

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
