## [Reviewer comments · Microbiology Spectrum]

Microbiology Spectrum

SarS and Rot are necessary for repression of *lukED* and *lukSF-PV* in *Staphylococcus aureus*

Exene Anderson, Juliana Ilmain, and Victor Torres

Corresponding Author(s): Victor Torres, St Jude Children's Research Hospital

Review Timeline:

Submission Date:	April 19, 2023
Editorial Decision:	May 15, 2023
Revision Received:	June 7, 2023
Editorial Decision:	June 15, 2023
Revision Received:	August 18, 2023
Accepted:	August 21, 2023

Editor: Jose Lemos

Reviewer(s): The reviewers have opted to remain anonymous.

Transaction Report:

DOI: <https://doi.org/10.1128/spectrum.01656-23>

May 15, 2023

Prof. Victor J. Torres
New York University Grossman School of Medicine
Microbiology
430 East 29th Street
Alexandria Center for Life Science, Rm 311
New York, NY 10016

Re: Spectrum01656-23 (SarS and Rot are both necessary for maximal repression of *lukED* and *lukSF-PV* in *Staphylococcus aureus*)

Dear Victor:

Thank you for submitting your manuscript to Microbiology Spectrum. On the basis of the reviewers' recommendations, I would like to receive a revised version of the manuscript that addresses the concerns raised by one or both reviewers. When submitting the revised version of your paper, please provide (1) point-by-point responses to the issues raised by the reviewers as file type "Response to Reviewers," not in your cover letter, and (2) a PDF file that indicates the changes from the original submission (by highlighting or underlining the changes) as file type "Marked Up Manuscript - For Review Only". Please use this link to submit your revised manuscript - we strongly recommend that you submit your paper within the next 60 days or reach out to me. Detailed instructions on submitting your revised paper are below.

Link Not Available

Sincerely,

Jose Lemos

Journals Department
Reviewer comments:

Reviewer #1 (Comments for the Author):

In the manuscript "SarS and Rot are both necessary for maximal repression of *lukED* and *lukSF-PV* in *Staphylococcus aureus*", Anderson and colleagues investigate repression of the *S. aureus* leucocidin genes by the regulatory proteins SarS and Rot. Collectively their data shown that both proteins play a role in repression of the toxin genes but that they do so in a non-additive manner. Although the study does not provide a detailed explanation for how these proteins work to co-repress the same genes the results are robust and the paper is excellently written. I have a couple of suggestions that the authors may want to address which are listed below.

Major issues

1. The title of the paper states that "SarS and Rot are both necessary for maximal repression of lukED and lukSF-PV in *Staphylococcus aureus*". I'm not sure the results presented support this conclusion. It would seem that EITHER protein is necessary, but not both since there appears to be little difference between either single mutant and the double mutant. Perhaps the authors could rephrase the title and other similar statements throughout the manuscript (e.g. on line 190 "We hypothesize that both repressors are needed to fully repress the toxins")

2. The DNA-Protein interaction experiments outlined in Fig 4A and 4B are suggestive of an interaction between the indicated promoters and SarS/Rot however a number of important controls are missing that would strengthen the experimental results. Typically EMSA experiments are performed using the same amount of DNA in each sample and a range of concentrations of the protein, however in Fig 4A only one concentration of each protein is used. Furthermore, it would be interesting to see how the DNA-Protein interaction is impacted if, for example, the concentration of one protein remains the same while the concentration of the second is increased. A necessary control for EMSA experiments is the addition of non-specific competitor DNA and protein. This is often accomplished by including poly dI-dC and BSA in the binding buffer, however no non-specific competitors are used here. In addition another common control is adding 100 X excess cold DNA to abrogate an apparent shift by releasing the bound labeled DNA fragment. Non-specific DNA, i.e. poly(dG:dC), was added to the promoter pull down experiments but no non-specific protein was used. Is it possible that the lukSG and lukED DNA fragments are simply binding non-specifically to any protein present in the sample.

Technical note - the authors may also want to consider adding a detergent like NP40 to the binding buffer as it can help improve resolution of shifted bands.

Reviewer #2 (Comments for the Author):

This manuscript builds upon previous data on SarS and Rot showing that they repress expression of leukocidins, bicomponent pore-forming toxins. The overall finding is that there is not a synergistic effect between SarS and Rot when both are absent in the bacteria. Overall, the data supports the major conclusion and provides data that will be of interest to the *S. aureus* research community. Accordingly, there is enthusiasm for the manuscript; however, the manuscript would be strengthened upon addressing several weaknesses noted below.

Major comments

There appears to be a strong case in this study that rot dominates over sarS and this should be pointed out. Figure 1 and text, as worded does not indicate that any differences between mutants was observed. The rot mutant led to increased lukED expression (both timepoints) compared to sarS and the rot sarS double mutant acted similar to rot. This would suggest that rot is dominant over sarS. This carries over into figure 2 and the associated text. For the immunoblots, rot seems to have a stronger change compared to sarS, at least at 3 hrs. Again, double mutant looks like rot. Figure 3c has this same trend.

Figure 3C, was there a statistically significant difference between sarS, rot, and sarS rot? From the data, it looks like the absence of rot in the sarS mutant rescues some of the killing.

Figure 4. These are not great looking EMSAs. Normally, one would titrate the protein. The other more pressing issue is that in both EMSAs there are two bands in the promoter only lane. The lower band makes perfect sense, but the upper band in each correlate to the size of one of the shifts. Panel B is much better.

Table 1 would benefit from more detailed description.

1) AH-LAC looks like it is AH1263 based on the reference. I am fine with calling it AH-LAC in the paper for shorthand or as a historical lab designation, but it should be pointed out in table

2) While in the methods, I would suggest making Table 1 basic genotype as written in the methods section. This would be helpful for a quick reference for reader. Then add a footer for spec, ermC, and ErmS denoting spectinomycin resistance, erythromycin resistance, and erythromycin susceptible

Minor comments

Do the strains grow similarly and does this impact the data especially considering the link to Agr? This is somewhat mitigated by the authors stating the phase of grown and not just time that the cultures were grown.

Line 169-170 and Figure 4C. "a significant decrease". There is no statistical analysis to support this statement. Also, the bars in

4C and 4D are not labeled or denoted in the text.

Line 94-95, 119, 122, 133, 135 should read wild-type.

Line 104,114 should read wild type,....

Use of wild-type and wildtype should be checked in figure legends and remaining text

Figure 3A,B labeling of WT and LAC is inconsistent

Figure 1B stats are surprising between sarS rot and sarS considering rot and sarS rot are significant. Also, not sure a 2-way ANOVA is appropriate.

In Figure 2A, sarS rot double mutant appears to have less intense bands than either rot or sarS alone, therefore could be different. Hard to tell from a single gel and lack of densitometry data.

Staff Comments:

Preparing Revision Guidelines

Please return the manuscript within 60 days; if you cannot complete the modification within this time period, please contact me. If you do not wish to modify the manuscript and prefer to submit it to another journal, please notify me of your decision immediately so that the manuscript may be formally withdrawn from consideration by Microbiology Spectrum.

Response to Reviewers for Spectrum01656-23

We would like to thank the editor for the opportunity to resubmit our manuscript and the reviewers for their thoughtful comments. Below we provide point-by-point responses (blue) to their comments.

Reviewer #1:

In the manuscript "SarS and Rot are both necessary for maximal repression of lukED and lukSF-PV in *Staphylococcus aureus*", Anderson and colleagues investigate repression of the *S. aureus* leukocidin genes by the regulatory proteins SarS and Rot. Collectively their data shown that both proteins play a role in repression of the toxin genes but that they do so in a non-additive manner. Although the study does not provide a detailed explanation for how these proteins work to co-repress the same genes the results are robust and the paper is excellently written. I have a couple of suggestions that the authors may want to address which are listed below.

Response: We thank the reviewer for the enthusiasm in our findings and for the constructive feedback. We have edited the text to make our points clearer.

Major issues

1. The title of the paper states that "SarS and Rot are both necessary for maximal repression of lukED and lukSF-PV in *Staphylococcus aureus*". I'm not sure the results presented support this conclusion. It would seem that EITHER protein is necessary, but not both since there appears to be little difference between either single mutant and the double mutant. Perhaps the authors could rephrase the title and other similar statements throughout the manuscript (e.g. on line 190 "We hypothesize that both repressors are needed to fully repress the toxins")

Response: We agree that the title may be misleading. Based on the data in Figure 1 and 2, we believe that both repressors are necessary for repression of toxins, as when one repressor is deleted, there is upregulation of leukocidin expression and production. We believe that either both proteins are needed for maximal repression, or that when one protein is not present, it is unable to activate expression of the other protein, therefore leading to lack of repression of the leukocidins. We have edited the text to make this more clear. We have modified the title to "**SarS and Rot are necessary for repression of *lukED* and *lukSF-PV* in *Staphylococcus aureus***".

2. The DNA-Protein interaction experiments outlined in Fig 4A and 4B are suggestive of an interaction between the indicated promoters and SarS/Rot however a number of important controls are missing that would strengthen the experimental results. Typically EMSA experiments are performed using the same amount of DNA in each sample and a range of concentrations of the protein, however in Fig 4A only one concentration of each protein is used. Furthermore, it would be interesting to see how the DNA-Protein interaction is impacted if, for example, the concentration of one protein remains the same while the concentration of the second is increased. A necessary control for EMSA

experiments is the addition of non-specific competitor DNA and protein. This is often accomplished by including poly dI-dC and BSA in the binding buffer, however no non-specific competitors are used here. In addition another common control is adding 100 X excess cold DNA to abrogate an apparent shift by releasing the bound labeled DNA fragment. Non-specific DNA, i.e. poly(dG:dC), was added to the promoter pull down experiments but no non-specific protein was used. Is it possible that the lukSG and lukED DNA fragments are simply binding non-specifically to any protein present in the sample.

Technical note - the authors may also want to consider adding a detergent like NP40 to the binding buffer as it can help improve resolution of shifted bands.

Response: Thank for all the great suggestion on how to improve the EMSAs. Our goal with these experiments was to test if we were able to show shift in the DNA migration upon addition of Rot and SarS. We utilized an establish protocol that our lab has used to show binding between Rot and leukocidin promoters (see references: PMC3416255 and PMC4127135). We then implemented the promoter pull-down experiments to specifically address the main concern of specificity raised by the reviewer. As shown in Figure 4b, we demonstrated that while SarS and Rot are able to bind concurrently to *lukED* and *lukSF-PV* promoter DNA, they do not bind to the negative control promoter (i.e., *purA* promoter DNA). Of note, while we would have liked to perform additional EMSAs to address the above recommendations, we were unable to do so as the first author of the paper has recently graduated and have left the lab. We hope that the combined results of Figure 4A and 4B are sufficient to establish that SarS and Rot are able to bind concurrently to leukocidin promoters. We thank the reviewer for the technical note, and will implement this suggestion in future work in the lab.

Reviewer #2:

This manuscript builds upon previous data on SarS and Rot showing that they repress expression of leukocidins, bicomponent pore-forming toxins. The overall finding is that there is not a synergistic effect between SarS and Rot when both are absent in the bacteria. Overall, the data supports the major conclusion and provides data that will be of interest to the *S. aureus* research community. Accordingly, there is enthusiasm for the manuscript; however, the manuscript would be strengthened upon addressing several weaknesses noted below.

Response: We thank the reviewer for their enthusiasm and insightful ideas. We have edited the manuscript to reflect the helpful comments provided by the reviewer.

Major comments

There appears to be a strong case in this study that rot dominates over sarS and this should be pointed out. Figure 1 and text, as worded does not indicate that any differences between mutants was observed. The rot mutant led to increased lukED expression (both timepoints) compared to sarS and the rot sarS double mutant acted similar to rot. This would suggest that rot is dominant over sarS. This carries over into

figure 2 and the associated text. For the immunoblots, rot seems to have a stronger change compared to sarS, at least at 3 hrs. Again, double mutant looks like rot. Figure 3c has this same trend.

Response: We have edited the text to emphasis this point more clearly.

Figure 3C, was there a statistically significant difference between sarS, rot, and sarS rot? From the data, it looks like the absence of rot in the sarS mutant rescues some of the killing.

Response: There was not a statistically significant difference between the *sarS*, *rot* and the *sarS rot* strains. We have added this statement to the text for clarification.

Figure 4. These are not great looking EMSAs. Normally, one would titrate the protein. The other more pressing issue is that in both EMSAs there are two bands in the promoter only lane. The lower band makes perfect sense, but the upper band in each correlate to the size of one of the shifts. Panel B is much better.

Response: Please see response to point 2 of Reviewer 1. We also noticed the two bands in the promoter only lane, and ran a DNA gel of the promoter fragments being used and confirm there was no contaminating DNA present. We hypothesize this may be primer dimer.

Table 1 would benefit from more detailed description.

1) AH-LAC looks like it is AH1263 based on the reference. I am fine with calling it AH-LAC in the paper for shorthand or as a historical lab designation, but it should be pointed out in table

2) While in the methods, I would suggest making Table 1 basic genotype as written in the methods section. This would be helpful for a quick reference for reader. Then add a footer for *spec*, *ermC*, and *ErmS* denoting spectinomycin resistance, erythromycin resistance, and erythromycin susceptible

Response: We have edited Table 1 to reflect these comments.

Minor comments

Do the strains grow similarly and does this impact the data especially considering the link to *Agr*? This is somewhat mitigated by the authors stating the phase of grown and not just time that the cultures were grown.

Response: We have observed no difference in how the strain grow, as measured by optical density.

Line 169-170 and Figure 4C. "a significant decrease". There is no statistical analysis to support this statement. Also, the bars in 4C and 4D are not labeled or denoted in the text.

Response: We have edited the text to address these comments.

Line 94-95, 119, 122, 133, 135 should read wild-type.

Line 104,114 should read wild type,....

Use of wild-type and wildtype should be checked in figure legends and remaining text

Response: Thank you for pointing this out, we have fixed this.

Figure 3A,B labeling of WT and LAC is inconsistent

Response: Thank you for pointing this out, we have fixed this.

Figure 1B stats are surprising between sarS rot and sarS considering rot and sarS rot are significant. Also, not sure a 2-way ANOVA is appropriate.

Response: Thank you for this comment and for catching that. We have reanalyzed these data using a 1-way ANOVA and adjusted the Figure to reflect the new statistical analysis. At both 3 and 5 hours, the *sarS rot* double deletion strain exhibited significant increased upregulation of *lukED*, and at 3 hours the *rot* single deletion strain also results in significantly upregulation of *lukED*. However, we still see no significant difference in upregulation of *lukED* between the double deletion and the *rot* single deletion, and no difference in upregulation between the three strains for either of the other toxins.

In Figure 2A, sarS rot double mutant appears to have less intense bands than either rot or sarS alone, therefore could be different. Hard to tell from a single gel and lack of densitometry data.

Response: Thank you for this comment and we appreciate this being pointed out. As we were interested specifically in leukocidin production, we performed western blots in Figure 2B that show that the double mutants are producing similar levels of leukocidins as the *rot* single deletion strain.

June 15, 2023

Prof. Victor J. Torres
New York University Grossman School of Medicine
Microbiology
430 East 29th Street
Alexandria Center for Life Science, Rm 311
New York, NY 10016

Re: Spectrum01656-23R1 (SarS and Rot are necessary for repression of *lukED* and *lukSF-PV* in *Staphylococcus aureus*)

Dear Victor,

Thank you for submitting your revised manuscript to Microbiology Spectrum. I am generally pleased with your revisions and appreciate the reasons for not performing additional EMSAs with protein titrations and proper controls as required by both reviewers. However, I don't believe the EMSAs shown in Figure 4A are of "publication quality". Not only because it lacks controls but also because there is not a defined/convincing band showing a shift or super-shift in the SarS or Rot+SarS lanes, respectively. With that said, I think the rest of the manuscript is sound and I am willing to accept a revised version without the EMSAs, such that the promoter pull-downs will serve as sole evidence that both proteins bind to *luk* promoters. Alternatively, I am happy to extend the deadline for submission of a revised version that will include the additional EMSAs. I also noted few minor issues in the manuscript that requires your attention:

1. Line 59. GdpS not gdpS
2. Line 100. 'increase upregulation'? Sentence requires revision.
3. Line 106. *lukED* in italics
4. Line 338. Indicate that $1 \times 10^{8.5}$ CFU was injected by i.p.
5. Line 397-401. One-way ANOVA for qRT-PCR

Link Not Available

Sincerely,

Jose Lemos

Journals Department
Reviewer comments:

Staff Comments:

Preparing Revision Guidelines

Please return the manuscript within 60 days; if you cannot complete the modification within this time period, please contact me. If you do not wish to modify the manuscript and prefer to submit it to another journal, please notify me of your decision immediately so that the manuscript may be formally withdrawn from consideration by Microbiology Spectrum.

Response to Reviewers for Spectrum01656-23

We would like to thank the editor for the opportunity to resubmit our manuscript. Please see below our point-by-point responses (blue).

Editor Comments

I am generally pleased with your revisions and appreciate the reasons for not performing additional EMSAs with protein titrations and proper controls as required by both reviewers. However, I don't believe the EMSAs shown in Figure 4A are of "publication quality". Not only because it lacks controls but also because there is not a defined/convincing band showing a shift or super-shift in the SarS or Rot+SarS lanes, respectively. With that said, I think the rest of the manuscript is sound and I am willing to accept a revised version without the EMSAs, such that the promoter pull-downs will serve as sole evidence that both proteins bind to luk promoters. Alternatively, I am happy to extend the deadline for submission of a revised version that will include the additional EMSAs.

Response: Thanks for your understanding. We agree with your assessment. To not delay the publication of this work any further, we have removed the EMSAs as we agree that the promoter pull-downs fully support our conclusions.

Minor issues

1. Line 59. GdpS not gdpS

Response: as *sarS* is being regulated by *gdpS* mRNA, we believe this should stay in the format it currently is.

2. Line 100. 'increase upregulation'? Sentence requires revision.

Response: We revised this sentence to read "There was increased upregulation of *lukED* in the double deletion strain as compared to the *sarS* deletion strain at both 3 and 5 hours"

3. Line 106. *lukED* in italics

Response: we italicized this

4. Line 338. Indicate that 1×10^8 CFU was injected by i.p.

Response: We included that the infection was performed by intraperitoneal injection

5. Line 397-401. One-way ANOVA for qRT-PCR

Response: Thank you for catching this- we have changed this to one-way ANOVA

August 21, 2023

Prof. Victor J. Torres
St Jude Children's Research Hospital
Host-Microbe Interactions
262 Danny Thomas Place, D2038D
Memphis, TN 38105

Re: Spectrum01656-23R2 (SarS and Rot are necessary for repression of *lukED* and *lukSF-PV* in *Staphylococcus aureus*)

Dear Victor:

Your manuscript has been accepted, and I am forwarding it to the ASM Journals Department for publication. You will be notified when your proofs are ready to be viewed.

Sincerely,

Jose Lemos
Editor, Microbiology Spectrum
